# Evaluation of an Innovative Care Pathway in the Diagnostic and Therapeutic Management of Hepatobiliary and Pancreatic Pathologies: “One-Day Diagnosis”

**DOI:** 10.3390/jpm13010012

**Published:** 2022-12-21

**Authors:** Zineb Cherkaoui, Barbara Seeliger, Vanina Faucher, Céline Biermann, Arne Kock, Patrick Pessaux

**Affiliations:** 1Department of Visceral and Digestive Surgery, Nouvel Hôpital Civil, University Hospitals of Strasbourg, 67000 Strasbourg, France; 2Group «Relevance and Care Pathways», Inserm Unit UMR_S1110, Institute for Research on Viral and Liver Disease, 67000 Strasbourg, France; 3SSPC (Simplification of Surgical Patient Care), UR UPJV 7518, Université de Picardie Jules Verne, 80000 Amiens, France; 4IHU-Strasbourg, Institute of Image-Guided Surgery, 67000 Strasbourg, France; 5Radiology Department, Nouvel Hôpital Civil, University Hospital of Strasbourg, 67000 Strasbourg, France; 6Anesthesiology Department, Nouvel Hôpital Civil, University Hospital of Strasbourg, 67000 Strasbourg, France

**Keywords:** one-day diagnosis, hepatobiliary and pancreatic diseases, care pathway, cancer, innovation, organization, value-based healthcare (VBHC)

## Abstract

“One-Day Diagnosis” (1DD) for hepatobiliary and pancreatic (HBP) diseases is an innovative care pathway that combines, on the same day, surgical consultation, medical imaging, anesthesia, diagnosis announcement, and therapeutic support consultations. The objective was to evaluate the length of the 1DD care pathway compared to a conventional one. The prospective “1DD care pathway” arm included 330 consecutive patients (January 2017–April 2019) vs. 152 (November 2014–November 2015) in the retrospective “conventional” one. In the 1DD group, diagnosis was made on the same day in 83% of consultations vs. 68.4% (*p* = 0.0005). Although there was no difference in overall time to diagnosis, diagnostic and therapeutic management was faster in the 1DD group (1 day vs. 15 days, *p* < 0.0004). In addition, 77% of patients who benefited from 1DD were very satisfied with their treatment overall. The mean cost of the 1DD consultation was EUR 176.8 +/− 149 (range: 50–546). The median cost of the overall program was similar (EUR 584 vs. EUR 563, *p* = 0.67). As an organizational innovation, the 1DD for HBP pathologies is a promising care pathway that optimizes diagnostic and therapeutic management, without creating medical overconsumption or additional costs. Given patient satisfaction, this model should be generalized to optimize cancer care by adapting it to the constraints of different healthcare structures.

## 1. Introduction

Liver cancers are the seventh most common cancer, with a prevalence of 994,539 cases worldwide and an incidence of 9.5/100,000 in 2020, and are the third-leading cause of cancer-related deaths (8.7/100,000) [1]. For hepatocellular carcinoma, the most common liver cancer [2], Singal et al. [3] reported a median time to diagnosis of 1.7 months, with a delay of more than 3 months in 31% of patients. After adjustment for tumor stage and Child–Pugh classification, failure to deploy all treatment options and delay in treatment were significantly associated with worse survival.

Pancreatic cancer is the twelfth most common cancer worldwide, with an annual incidence of 4.9/100,000, and the seventh leading cause of cancer-related deaths (4.5/100,000) in 2020 [1], with a 5-year survival rate of only 9% [4]. The prognosis of pancreatic cancer depends particularly on the time to diagnosis [5]. The REPERE national survey on care pathways for patients with metastatic pancreatic cancer in current practice in France reports an estimated median delay between the onset of first symptoms and diagnosis ranging from 41 to 65 days [6].

In France, liver and pancreatic cancers account for 13% and 18%, respectively, of all cancers of the digestive tract; they are the second and third most common in men, and the third and second most common in women. Liver and pancreatic cancers are the fourth and fifth most common causes of cancer-related death in men and seventh and fourth in women, respectively [7]. The initially silent evolution, the aggressiveness of these cancers, and their poor prognosis require early diagnosis in order to use all possible therapeutic means. Against this background, the first objective of the Cancer Plan 2014–2019 [7] was to promote earlier diagnosis by identifying new opportunities for the earliest possible diagnosis.

In this context, the organizational innovation of “One-Day Diagnosis” (1DD) for hepatobiliary and pancreatic (HBP) diseases was initiated [8]. The rationale of the study was to bring together all actors and resources needed for diagnosis and initiation of the treatment pathway within one day. Based on the experience in senology [9], accelerated diagnosis and treatment is crucial, especially for diseases with poor prognoses that represent health management challenges.

The 1DD for HBP diseases is an innovative care pathway that combines, on the same day, surgical consultation, medical imaging, anesthesia, diagnosis announcement, and therapeutic support consultations. The aim of this work was to analyze the outcomes of patients who had consecutively benefited from a 1DD compared to historical patients who had followed a “classic” pathway. The primary objective was to measure the time taken to complete the care pathway to diagnosis and treatment, and the secondary objectives were to analyze the medico-economic impact, the impact in terms of care pathways, patient satisfaction, and the evolution of all these outcomes between the first six months and the last six months of implementation of this accelerated pathway.

## 2. Materials and Methods

### 2.1. Cohort

This noninterventional, monocenter study compared two groups of patients referred for diseases of the liver, biliary tract, and/or pancreas. 

The “1DD care pathway” study arm, whose operational description has been reported [8], included prospective data collection for all patients consecutively managed between January 2017 and April 2019. The second arm was a retrospective control group, referred to as the “conventional care pathway”, which included all consecutive patients taken care of between November 2014 and November 2015. This period was chosen to avoid the year between the two care pathways when adaptations of 1DD implementation may have occurred. All individuals seen for follow-up consultations or for other digestive pathologies during both periods were excluded from the study (Figure 1).

Prospective enrollment of patients in the 1DD arm was based on information and explanation provided by the surgeon during the initial consultation, and a signed informed consent form. For the “conventional pathway”, all consecutive consultation appointments recorded in the institutional software were collected thoroughly and chronologically.

### 2.2. Data Collected

The following data were collected based on the medical record: Sociodemographic: age, sex, and type of pathology.Organizational: in relation to the stages of the treatment pathway before the initial surgical consultation: date of consultation with the referring physician, his/her specialty and place of practice, and complementary examinations performed with their date and place.Clinical (from the surgical consultation to the diagnosis and treatment): date of surgical consultation, reason for consultation, date of appointment for “1DD”, complementary examination(s) requested or performed on the same day, date and location of these examinations, type and date of diagnosis, date of announcement consultation (if applicable), date of therapeutic accompaniment consultation (if indicated), type of treatment proposed, and date of the anesthesia consultation (if indicated for surgery).Medico-economic: recording of all costs according to the tariff of the primary health insurance.

For cancers, treatment decisions were systematically validated in a multidisciplinary team meeting.

At the end of the 1DD day, each patient completed a satisfaction questionnaire that assessed the clarity of the information and instructions received beforehand; the ease of finding where the 1DD was to take place; the quality and clarity of the exchanges with the reception staff and the secretariat; the waiting time before the treatment; the quality and clarity of the exchanges with the surgeon, medical imaging staff, radiologist, anesthesiologist, and therapeutic support nurse; the overall duration; the respect of confidentiality; and the overall appreciation. In turn, the impressions and feedback of the medical staff were collected in open interviews.

### 2.3. Definitions

Time of pathway completion was measured in days, from the first complementary examination, marking entry into the pathway, until the treatment. 

### 2.4. Statistical Analyses

Quantitative variables were described with standard position and dispersion statistics, namely, mean, median, variance, minimum, maximum, and quantiles. The delay variables were asymmetrically distributed and were represented by their median and quantiles [q2.5; q97.5]. Qualitative variables were described by the proportions of occurrence of the modality.

The Gaussian nature of the quantitative variables was assessed using the Shapiro–Wilk test. 

For the comparison of a quantitative variable between two groups, Student’s *t*-test or Mann–Whitney–Wilcoxon test was used, depending on the hypotheses for the application of each test. For the analysis of qualitative variables, the parametric chi-squared test was used when the conditions of application permitted, or the Fisher’s exact test was used when they did not. 

The alpha risk was set at 5% in all analyses, and a value of *p* < 0.05 was considered significant.

All analyses were performed using R software version 3.1.0, which is equipped with all additional libraries required for data analysis (R Core Team (2014). R: A language and environment for statistical computing. R Foundation for Statistical Computing, Vienna, Austria).

## 3. Results

### 3.1. Cohort

The study cohort initially consisted of 708 patients, divided into two groups: 375 patients in the 1DD arm and 333 patients in the conventional care pathway arm. After exclusion of follow-up visits and visits for digestive diseases other than HBP, 330 and 152 patients were analyzed, respectively (Figure 1).

The patients were similar in the two study arms (Table 1). The distribution of the reasons for consultation was similar (*p* = 0.06). The four other pathologies encountered during the 1DD were the following: gastric adenocarcinoma invading the pancreas; suspected pancreatic lesion not objectified after imaging; aneurysm of the pancreaticoduodenal artery.

### 3.2. Analysis between the Two Groups

In the 1DD group, the diagnosis was made on the same day in 83% of visits versus 68.4% in the conventional care pathway group (*p* = 0.0005). This difference was statistically significant for malignant diseases (84.8% in the 1DD arm compared with 61.3% in the standard care pathway arm, *p* = 0.0001). The 1DD consultation was scheduled within a median of 14 days of making the appointment. Patients in the 1DD arm had undergone fewer additional examinations before the surgical consultation: 94.2% versus 99.3%, *p* = 0.006 (Table 2). On the 1DD appointment, 18.5% of patients had a CT scan on the same day; 30% had a magnetic resonance imaging (MRI) scan; and 8.5% had both (CT + MRI).

The median time to complete the overall pathway was 69 days for the 1DD versus 65.5 days for the “conventional pathway” (*p* = 0.69). The median time from diagnosis to initiation of appropriate treatment was 1 day versus 15.5 days (*p* = 0.0003), regardless of whether the diagnosed condition was benign or malignant (Table 3). Once the diagnosis was established, there was no significant difference in the time from diagnosis to surgery (34 versus 35 days; *p* = 0.69) (Table 3).

The time from diagnosis to anesthesia consultation when surgery was indicated was 1 day in the 1DD arm versus 23.5 days in the conventional pathway arm (*p* < 0.001), with an anesthesia consultation on the same day as the surgical consultation in 45.9% of the 1DD group. The delay between diagnosis and surgery for patients in the 1DD group with cancer was 52 days in the absence of a same-day anesthesia consultation versus 23 days when it was performed on the same day (*p* < 0.001). Similarly, for benign pathologies requiring surgery, the delay was 55.5 days versus 27 days (*p* = 0.007). The number of consultations was three in the 1DD pathway versus four in the conventional pathway (*p* = 0.003).

The implementation of this innovative pathway changed the profile of referring physicians, with an increase from 52.4 to 74.3% of patients being referred by physicians outside the University Hospital Center (*p* < 0.0001). Before the 1DD, 48.1% of patients had initially consulted a gastroenterologist, whereas after its implementation, 31% of patients were referred directly by their general practitioners (*p* = 0.04).

### 3.3. Analysis of the First 6 Months and the Last 6 Months in the “1DD Care Pathway” Group

During the first 6 months, 55 patients had benefited from this innovative pathway compared with 99 patients during the last 6 months. The two groups were similar (Table 4), but with more diagnoses of benign pathology in the last 6 months.

The median duration of the overall pathway was 77.5 days in the “first 6 months” group versus 60 days in the “last 6 months” group (*p* = 0.09). The median time from diagnosis to initiation of treatment was significantly shorter in the last 6 months (19 days versus 1 day, *p* = 0.01) (Table 5). This significant difference amounted to 23 versus 9 days for the management of cancer (*p* = 0.03). The number of journeys was reduced from three to two (*p* = 0.01).

Between the first and last 6 months, there was no local difference in the place of practice of the referring physicians (outside or within the university hospital). The number of patients seen by a gastroenterologist dropped from 47.6% to 28.9%, as there were significantly more patients directly referred by their general practitioner (increase from 14.3% to 39.8%, *p* = 0.04).

### 3.4. Satisfaction of the Study Population

This accelerated care pathway was rated as satisfactory by 94% of patients and excellent by 77%. After the open-ended interviews, 19 healthcare professionals were convinced of the benefits of this new accelerated treatment and wished to pursue this pathway and extend it to other oncological diseases.

### 3.5. Medico-Economic Analysis

The mean cost of the 1DD consultation was EUR 176.8 +/− 149 (range: 50–546). The median cost of the overall treatment pathway (from the first complementary examination to treatment) was EUR 584 in the 1DD group, compared with EUR 563 in the conventional pathway group, with no statistically significant difference (*p* = 0.67).

## 4. Discussion

The accelerated diagnostic and therapeutic management pathway for HBP pathologies through the 1DD model allowed the diagnosis to be established in 83% of cases on the day of the consultation, which is particularly relevant for malignant pathologies. The delay between diagnosis and the implementation of appropriate treatment was significantly reduced. The duration of the overall care pathway has not been reduced, although a trend is beginning to emerge over the last 6 months of implementation of the system, raising questions about possible levers upstream of the 1DD consultation. This accelerated care was considered satisfactory by 94% of patients, with 77% considering it excellent.

For one-day diagnosis of breast cancer, an accurate diagnosis was made within the same day in 75% of cases [9]. The rate in this series for HBP pathologies is higher, probably due to the fact that the diagnosis of liver tumors (benign or malignant) is essentially based on imaging, with therapeutic management of hepatocellular carcinoma without the need for histological proof [10]. In cases where the diagnosis could not be established, an interventional procedure was required in most cases to obtain a histologically confirmed diagnosis, such as liver puncture biopsy and endoscopic ultrasound with pancreatic biopsy.

This pathway has reduced the number of complementary examinations prior to surgical consultation as well as their redundancy. Moreover, the provision of imaging through this organizational innovation does not imply multiplication and excessive consumption of examinations. This is reflected in the numbers of examinations performed on 1DD, as less than one-third of patients require them. It should be noted that the review of images by an experienced radiologist in consultation with the HBP surgeon on the same day avoids repetition of imaging exams. The same location in our model greatly favors this close collaboration and should be considered when implementing such a pathway.

The reduction of the delay between diagnosis and therapeutic decision-making meets the objective of early management of cancer diseases in order to improve survival. In our analysis, we did not find any difference between the groups in terms of the overall length of the care pathway, although a trend emerged in the last 6 months of implementation of the system (60 days versus 77.5 days in the “first 6 months” group; *p* = 0.09). The effect of the 1DD organizational innovation on entry into the care pathway is not a direct one but relies on the territorial care network organization. Collaboration and coordination between health professionals promote appropriate referrals and reduce delays in accessing the 1DD consultation, thus limiting healthcare travel and the multiplication of complementary tests. Digital tools can probably be used to facilitate the operational deployment of this territorial network. Raising awareness among general practitioners is particularly important in the process of accelerating diagnostic management [6]. Raptis et al. suggested that early referral to a specialized unit for pancreatic pathologies would be one of the most effective measures for curative management and, thus, improved survival [11]. The earlier pancreatic cancer is detected and treated, the sooner curative surgery improving prognosis can be offered.

Despite the methodological limitations of the available studies, the clinical benefit of reducing delays in care for patients with pancreatic cancer is supported by several studies (including 8 of 19 with multicenter analyses), particularly in patients with potentially curable disease [12]. Only one study [3] reported the risks of delayed therapeutic management of hepatocellular carcinoma. In particular, delayed treatment was associated with more frequent ascitic decompensation (hazard ratio = 2.8; 95% CI 1.3–6.1). After adjustment for tumor stage and degree of cirrhosis according to the Child–Pugh classification, delayed treatment was one of the factors leading to worse survival (HR = 0.50; 95% CI 0.30–0.84). A literature review on the correlation between speed of diagnosis and clinical outcome of various cancers showed that the earlier the diagnosis, the lower the morbidity and mortality [13]. Thus, a model such as the 1DD should be evaluated for all cancer diseases [13,14,15].

Patients within the 1DD reported satisfaction with this rapid management. This reduces the anxiety of waiting for results before the diagnosis is announced [16]. Berman et al. [17] evaluated the satisfaction of patients who had received a one-day diagnosis of breast lesions and the anxiety generated by the prediagnosis phase. They reported that “Even though faster access to treatment does not reduce the psychological morbidity of awaiting diagnosis, the patients express their satisfaction and find the rapidity of the pre-diagnosis phase beneficial” [17]. A consultation with a psychologist could also be organized on the same day to alleviate this anxiety as much as possible, even if its duration is limited by the 1DD [10]. In this interval, the accompaniment of the patient by a pathway coordinator seems essential throughout the day. On the other hand, this pathway places the patient back at the center with health professionals attentive to his or her wellbeing in parallel with their care activities. Waiting times for the patient between different health appointments and the anxiety generated are then reduced, favoring complete, global, and quality care, offering a wide range of therapeutic options. Moreover, 77% of patients benefiting from this new pathway say they are very satisfied with their overall care.

These organizational innovations are difficult to evaluate. We are evolving in a system that is inflationary by nature, valuing only the quantity of medical acts, at the cost of rewarding quality and especially relevance. Today, surgery has few quality indicators and certainly no indicators of relevance of care. A relevant care pathway requires quality procedures, but this is not enough. Relevance of care includes quality, but it also integrates notions of organization, indication, follow-up, and coordination of the different care providers. In industry or services, quality assurance has long been commonly used to meet the desired level of quality. In addition to assessing the quality of the care provided according to the Patient-Reported Experience Measurements (PREMs) and Clinical Reported Outcomes Measurements (CROMs), patients must also be asked about their quality of life over the long term, in order to verify that we are providing results that really matter to them in their daily lives (Patient-Reported Outcomes Measurements, PROMs). It is this multidimensional approach that will allow a true evaluation of practices and the value of the care provided [18]. These standardized data, shared transparently between teams and accessible to patients, will also be a robust tool for evaluation and improvement [19]. Moreover, all these data become a source of information managed in real time by the practitioner in order to adapt and personalize the management.

Our study has some limitations. It is a monocentric study taking place within the university hospital Institute of Image-Guided Surgery, a model developed within clinical research. In order to promote the development and generalization of this model of accelerated care, it is now necessary to implement it in other centers in order to carry out comparative studies and to confront the different operational obstacles. On the other hand, data collection for the control group was performed retrospectively, with difficulties in tracing all the elements relating to the management of the course prior to the first consultation from the data in the medical records. For example, it was difficult to ascertain the exact number of medical consultations prior to our care, which could lead to a bias against the 1DD group for the calculation of costs. Similarly, for the medico-economic evaluation, it was not possible to include the costs of healthcare travel, as we did not know their price (type, fixed mileage), again to the disadvantage of the 1DD group, because of the lower average number of journeys in this group. 

The present study underlines the central role of such an optimized care pathway, which accelerates the initiation of treatment, particularly for rapidly evolving cancer pathologies and/or those with a poor prognosis. There is, therefore, a definite interest in its dissemination to all oncology consultations. Today, cancer pathologies represent a public health challenge, and health professionals aim to diagnose them as early as possible in order to deploy all possible curative treatment options at the time of diagnosis while ensuring the quality of life of patients. 

The 1DD model, as an organizational innovation, is a promising care pathway, optimizing diagnostic and therapeutic management, without creating medical overconsumption or additional costs. In view of the patients’ satisfaction, this model deserves to be disseminated and generalized while adapting it to the constraints of the different healthcare structures.

## Figures and Tables

**Figure 1 jpm-13-00012-f001:**
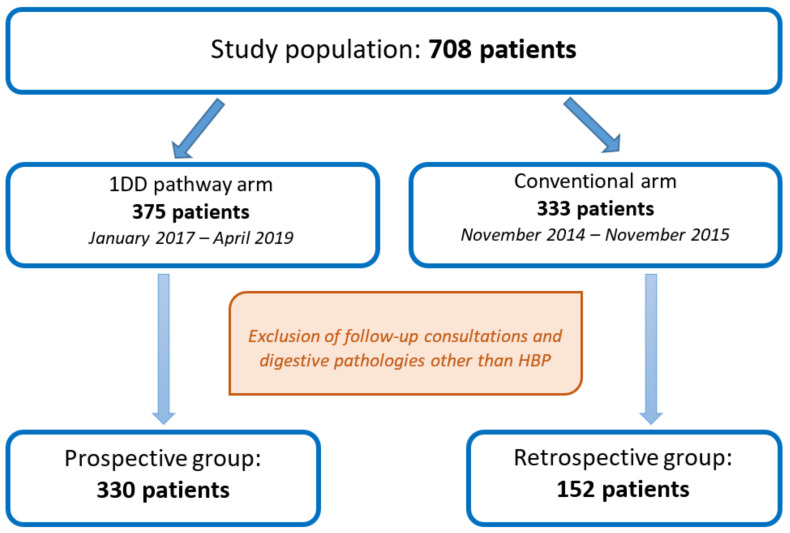
Study flowchart.

**Table 1 jpm-13-00012-t001:** Study group characteristics in both pathways.

Characteristics	1DD Pathway	Conventional Pathway	*p*
Age in years (median, range)	67 (25–89)	65 (28–88)	0.62
Sex F/M (%)	51.2/48.8	49.3/50.7	0.77
Benign pathologies (%)	52.1	50.7	0.77
Malignant pathologies (%)	47.9	49.3	
**Types of pathologies**			0.06
Hepatic (%)	40.6	48	
Hepatic and biliary (%)	0.3	0	
Hepatic and pancreatic (%)	1.5	3.3	
Pancreatic (%)	37	37.5	
Biliary (%)	17	11.2	
Duodenal (%)	2.4	0	
Other (%)	1.2	0	
**ASA classification * (%)**			0.35
ASA 1	8.2	5.9	
ASA 2	60.7	51	
ASA 3	29.5	43.1	
ASA 4	1.6	0	

* ASA (American Society of Anesthesiologists) score of patients who underwent surgery.

**Table 2 jpm-13-00012-t002:** Comparisons between study groups at the time of the first HBP surgeon consult.

	1DD Pathway	Conventional Pathway	*p*
**Diagnosis made at first HBP surgeon consult (%)**	**83**	**68.4**	**0.0005**
Subgroup “malignant pathologies”	84.8	61.3	0.0001
Subgroup “benign pathologies”	81.4	75.3	0.31
**Previous complementary examinations (%)**	**94.2**	**99.3**	**0.006**
Subgroup “malignant pathologies”	93.7	100	0.03
Subgroup “benign pathologies”	94.8	98.7	0.18

**Table 3 jpm-13-00012-t003:** Comparison of time to care between study groups.

Delay (in Days)	1DD Pathway			Conventional Pathway			*p*
	Median	Quantile 2.5	Quantile 97.5	Median	Quantile 2.5	Quantile 97.5	
**Pathway duration between first complementary examination and treatment**	**69**	**6**	**412**	**65.5**	**6.35**	**334.10**	**0.69**
Subgroup “malignant pathologies”	68	5.85	378.35	59	7.10	203.75	0.35
Subgroup “benign pathologies”	69.50	6	438.25	70	13.60	358	0.72
**Time from diagnosis to treatment**	**1**	**1**	**127.65**	**15.5**	**1**	**254.35**	**0.0003**
Subgroup “malignant pathologies”	16	−8.70	137.80	21	1	150.40	0.02
Subgroup “benign pathologies”	1	1	95.88	1	1	395.70	0.01
**Surgical patients**							
Time from diagnosis to anesthesia consultation	1	1	20.50	23.50	1	150.35	<0.001
Time from diagnosis to surgery **	34.00	10.85	204.25	35.00	5.25	246.20	0.69

** Excluding surgeries scheduled at the patients’ convenience.

**Table 4 jpm-13-00012-t004:** Subgroup characteristics in the 1DD care pathway arm.

Subgroup Characteristics	First 6 Months	Last 6 Months	*p*
Age in years (median, range)	70 (28–82)	66 (25–88)	0.53
Sex F/M (%)	49.1/50.9	54.5/45.5	0.61
Benign pathologies (%)	40	66.7	0.002
Malignant pathologies (%)	60	33.3	
**Types of pathologies**			0.23
Hepatic (%)	47.3	34.3	
Hepatic and biliary (%)	1.8	0	
Hepatic and pancreatic (%)	1.8	2	
Pancreatic (%)	34.5	35.4	
Biliary (%)	12.7	24.2	
Duodenal (%)	1.8	1	
Other (%)	0	3	
**ASA classification * (%)**			0.27
ASA 1	0	18.2	
ASA 2	57.9	54.5	
ASA 3	36.8	27.3	
ASA 4	5.3	0	
**Diagnosis made at first HBP surgeon consult (%)**	83.6	79.8	0.67
**Previous complementary examinations (%)**	89.1	95	0.20

* ASA (American Society of Anesthesiologists) score of patients who underwent surgery.

**Table 5 jpm-13-00012-t005:** Comparison of care pathway delays between subgroups in the 1DD care pathway arm.

Delay	First 6 Months			Last 6 Months			*p*
	Median	Quantile 2.5	Quantile 97.5	Median	Quantile 2.5	Quantile 97.5	
**Pathway duration between first complementary examination and treatment**	**77.50**	**13.95**	**259**	**60**	**7.20**	**281**	**0.09**
Subgroup “malignant pathologies”	70.50	18.20	182.80	57	12.20	183.80	0.29
Subgroup “benign pathologies”	106.50	11.35	308.05	61	4.30	300.18	0.08
**Time from diagnosis to treatment**	**19**	**−4.55**	**146.60**	**1**	**1**	**85.75**	**0.01**
Subgroup “malignant pathologies”	23	−7.20	144.60	9	−3	65.60	0.03
Subgroup “benign pathologies”	1	1	110.38	1	1	81.38	0.43
**Number of journeys prior to the HBP surgeon consult**	**3**	**1**	**5**	**2**	**1**	**4**	**0.009**
Subgroup “malignant pathologies”	3	1	5.55	3	1	4	0.26
Subgroup “benign pathologies”	3	1	4	2	1	4	0.046

## Data Availability

The data presented in this study are available on request from the corresponding author. The data are not publicly available due to privacy or ethical restrictions.

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
