# Peer review of "Evaluation of an Innovative Care Pathway in the Diagnostic and Therapeutic Management of Hepatobiliary and Pancreatic Pathologies: “One-Day Diagnosis”"

_jpm, 2022, doi:10.3390/jpm13010012_

Round 1
Reviewer 1 Report
Introduction: 905677 new cases – the proper display should be XXX per 1000 persons or one million persons – so readers will know the nominator and the denominator of liver cancers incidence and deaths rate .
Table 1 -5 - Numbers and P-value decimal point should be unified. Some was no decimal point, some was 1 or 2 decimal point – need to be unified. Continuous variables should report mean +/- SD and categorical variables should report N and (%).
2.4. Statistical Analyses – all methods were univariable methods – should also analyze results multivariable way.
Author Response
Thank you for the time invested to thoroughly review our manuscript.
We modified the representation of prevalence, incidence, and mortality according to your suggestions.
We unified the p values to 2 significant digits according to your recommendation.
According to the discussion with our statistician involved in the data analysis of this study, median and range best represent the non-Gaussian data distribution.
The present study did not analyse risk factors for which a multivariate analysis could have been used. Thus, we performed univariate analysis.
Reviewer 2 Report
The manuscript describes the importance of 1Day diagnosis over the traditional/conventional way. the analysis was collectively performed on almost 480 patients in both groups. Overall the results are promising in favor of 1DD in cancer diagnosis and in prognosis too.
The introduction needs a few modifications regarding the background/rationale of the study.
Author Response
We are grateful for the time taken to review our manuscript and the encouraging comments.
The introduction was modified according to your suggestion to clarify the rationale of the study.